# Quality and Nutritional Parameters of Food in Agri-Food Production Systems

**DOI:** 10.3390/foods12020351

**Published:** 2023-01-11

**Authors:** Songül Çakmakçı, Ramazan Çakmakçı

**Affiliations:** 1Department of Food Engineering, Faculty of Agriculture, Atatürk University, Erzurum 25240, Turkey; 2Department of Field Crops, Faculty of Agriculture, Çanakkale Onsekiz Mart University, Çanakkale 17100, Turkey

**Keywords:** organic and traditional foods, nutritional value, food quality, secondary metabolites, antioxidants, phenolics, food safety, farming systems

## Abstract

Organic farming is a production system that avoids or largely excludes the use of synthetic agricultural inputs such as pesticides, growth regulators, highly soluble mineral fertilisers, supplements, preservatives, flavouring, aromatic substances and genetically modified organisms, and their products. This system aims to maintain and increase soil fertility and quality, and relies on systems such as crop rotation, polyculture, intercropping, ecosystem management, covering crops, legumes, organic and bio-fertilisers, mechanical cultivation and biological control methods. The present review summarises and evaluates research comparing the quality of traditionally, organically and conventionally produced foods. In some cases, although the results of the studies contradict each other, organically grown in vegetables, especially berries and fruits are slightly higher dry matter, minerals such as P, Ca, Mg, Fe and Zn, vitamin C, sugars, carotenoids, antioxidant activity, phenolic and flavonoid compounds. In addition, their sensory properties are more pleasant. The nutritional content, quality and safety of organic foods are acceptable if the recent trends are reviewed, tested and verified. Therefore, the aim of this review is to compile, describe and update scientific evidence and data on the quality, safety, bioactive compounds and nutritional and phytochemical quality of foods in traditional and organic fruit, vegetable and cereal production systems.

## 1. Introduction

Organic farming is a production management system that encourages and improves soil organic matter, biochemical and ecological characteristics, agricultural ecosystem health, biodiversity, natural biological cycles, soil biological activity and microbial richness, and enhances soil fertility by minimising the application of external inputs and maximising the efficient use of local resources [1,2]. Organic agriculture, which is a sustainable, agroecological, economic and holistic approach that improves soil health and related microbial communities, and provides environmental sustainability and is cost-effective, may be preferred over conventional methods for suitable regions, conditions and plants [3]. The soil, plant, environmental and biodiversity benefits of organic agriculture are mostly accepted, and its effects on foods’ nutritional composition are controversial [4]. Processed organic foods, compared with conventionally produced foods, contain much fewer synthetic additives, and no artificial colouring and flavouring agents, stabilisers, sweeteners, synthetic preservatives or aromatic substances [5]. The yield in organic systems lower than conventional ones, but organic foods have significantly less or no synthetic pesticide residues. This is especially important in children nutrition [6]. In addition, it is a fact that there are differences in manufacturing, processing, preparation, packaging, preservation, and marketing methods that can affect the quality of organic and conventional foods.

Comparisons of organic and conventional farming systems mostly focus on production, the environment, the economy, welfare and sustainability. Organic agriculture targets social, cultural, economic and ecological sustainability and better balances economic viability; moreover, it is more profitable and environmentally friendly Zhu et al. [7] determined that an organic production system could help reduce environmental impacts despite having lower productivity, while Clark [8] concluded that compared to conventional agriculture, organic farming is more effective at using non-renewable energy, and performs well in terms of energy efficiency. In addition, organic farming practices are thought to help contribute to rural development and the economy [6]; can be very beneficial for degraded and marginal land areas [9]; could be a solution to reducing the negative impact of conventional agriculture on the environment and natural resources [2,10]; increase soil microbial activities and bacterial diversity [11,12]; and improve climate-smart agriculture [13], soil quality indicators, acid and alkaline phosphatase activities [14] and the physico-chemical properties of soil [15]. Long-term organic management has been shown to lead to significant changes in the chemical and biological properties of soil, and has a long-term positive effect on soil quality and microbial diversity [12]. While organic agriculture continues to grow in response to consumer preferences as a part of future agriculture due to its beneficial practices, there are limitations to its widespread use, such as the need for more land and labour and low yields compared to conventional practices [2,16].

Lower yields, high food prices, difficult access, a lack of availability and irregular supply are the main criticisms generally reported for organic farming systems, although these vary depending on the crops, agroecological conditions and practices. There is scientific evidence that the yields under organic farming systems are approximately 20–25% lower than in conventional farming systems with high nitrogen fertiliser input [11,17]. However, the yield difference between organic and conventional agriculture may decrease over time, as organic agriculture will have a higher organic matter concentration, greater stability of soil properties, and improved soil structure with higher soil aggregation [16]. It has been shown that organic management at equal N levels can have an advantage over the conventional system in both the yield and quality of carrots [18]. Contrary to the usual expectation, orchards with organic management based on agroecological principles and input substitution have been shown to be efficient production systems, with the highest blueberry yield and lowest production cost compared to conventional orchards [19]. In a recent meta-analysis examining temporal yield stability in horticultural crops, no difference was found between organic and conventional management [10], while another general evaluation measuring the yield stability and variability of these systems showed that yield stability did not differ, or was lower in organic management [17]. 

Studies reveal that consumers generally have favourable expectations towards organic foods and buy these foods because they avoid pesticides (and other chemicals used in food production) and genetically modified foods; moreover, they believe them to be more nutritious, environmentally friendly, natural, healthy, safe, tasty, clean and high-quality [13,20]. While the effects of the agro-food industry on the environment and nutrition lead consumers to organic foods, food availability and affordability can also encourage the purchase of non-organic foods [21]. Several studies reveal that some of the prominent motivating factors for buying organic food are health-promotion or nutrition, attractiveness, nutritional and biological value, domain-specific innovativeness, origin, regionalism, health benefits and awareness, product quality, health and environmental consciousness, social and self-identity, food safety, trust, freshness, credibility, emotions, perception, and sensory properties such as taste, appearance, odour, flavour, intensive aroma, mouth feel and texture [22,23,24,25,26].

## 2. Organic Food

Organic food has been described as “food guaranteed to have been produced, stored, and processed without adding synthetic fertilisers and chemicals”. Organic foods and products are made from organically produced ingredients that are processed primarily via biological, mechanical and physical means. Organic foods differ from conventional ones, predominantly because of the absence of pesticide, artificial fertiliser and heavy metal residues due to the application of regulated production rules; the majority of scientific studies deal with the quality of organic food and these compounds in order to verify their limits [27]. Organic foods are naturally grown and produced via standard methods of organic agriculture, and their production relies on ecological processes, biodiversity and natural cycles.

While the flavour, quality and safety of traditional local foods are usually based on organic raw materials, organic agriculture systems seek to provide the consumer with natural, fresh, healthy, delicious, nutritious, health-promoting, more environmentally and biodiversity-friendly and authentic food while respecting natural life-cycle systems. Indeed, the perception that organic foods are eco-friendly, trustworthy, nutritious and healthier than conventional foods has led to an increase in their demand due to these safer, traceable and better-controlled foods products [13,28]. Organically grown plants and plant products contain fewer hazardous heavy metals, nitrates, nitrites, nitrogen and pesticide residues, and in contrast, they have more dry-matter content, sugars, ascorbic acid, total phenolic compounds and mineral nutrient content [18,27]. Soil nitrogen appears to affect nutritional quality, and organic crops generally have higher dry matter, sugar, ascorbic acid, flavonoids and phenolics, as well as lower yields and moisture, nitrate and protein content. The literature shows that organic grains, vegetables and fruits have higher or equal amounts of minerals, phytochemicals and vitamins C and E, while organic cereal has less protein, but an equal amount in vegetables and fruits [5]. It has been suggested that organic pasta has higher fibre and lower protein content than conventional pasta [29], while organic rice and wheat contain less protein and amino acids. Although there are nutritional differences between production systems, organic practices could produce safer rice and wheat [30]. The fertilization regime has a clear impact on protein composition. Due to low nitrogen availability in an organic system, especially in the reproductive phases, it has been found that protein content and gluten quality are low in wheat grain [31]. However, when the fertility level is similar, it has been determined that the protein content of whole wheat and refined flour is the same in organic and conventional systems, and that mineral and antioxidant content and wheat quality are more strongly associated with fertility level [32]. Indeed, in a comparison of three ancient wheat species, in terms of the quality characteristics of organic and conventional cultivation under equivalent nitrogen fertilization, Fares et al. [33] found that organic cultivation did not affect the phenolic acid profile and antioxidant activity, except to increase the total phenolic content.

## 3. Traditional Food

Nowadays, there is growing consumer interest in local, regional-origin products that have a traditional and natural character; authentic recipe processes; a regional identity, flavour, taste or image; sensory quality; and positive image, and are perceived as more sustainable and high-quality [34,35]. The stages of the production, processing and preparation of traditional foods are carried out in a certain area, and their recipes, the origin of their raw materials, and their production processes are authentic. Traditional foods have certain attributes or properties that clearly distinguish them from other similar products in the same category; for example, they might have traditional ingredients, a traditional composition, specific traditional raw materials or primary products, an authentic recipe that has been known for a long-time, traditional methods of production or processes [34]. Traditional foods have played a historically important role in different cultures and regions. They contribute to consumers’ sense of identity and pride. Consumers believe that traditional foods are fresh, natural and have a stronger more special taste, are nutritious, healthy and safe, and have higher nutritional value and higher quality. They describe them as homemade and natural. Therefore, the demand for traditional foods is increasing day by day [36]. Previous research emphasises that such products are considered a symbol of cultural values, and in transition economies, the choice of traditional foods is regarded as a psychological tool, and helps consumers to be related to trends in foods [34]. Traditional food products are often the result of cultural practices that protect and improve rural ecosystems, and their production contributes to rural development, sustainability and the conservation of nature and biodiversity.

The production of traditional foods helps the environment and diversifies agricultural activities, promoting regions and tourism, and following organic food trends [35]. Indeed, local, authentic and traditional foods that offer unique and memorable food and beverage experiences have been found to promote tourism through the creation or revival of cultural identity [37]. It has been considered that traditional-type foods are healthier and more organic, have a more intensive aroma and local nature, and are tastier than those that are mass-produced [23]. Traditional foods constitute an important form of economic input in the food sector, and create new income opportunities for farmers in an ecosystem of a traditionally systemised countryside food style that is raw and simple, local, artisanal, healthy and organic. The sensory properties and quality of traditional foods are generally based on the source of organic raw materials. On the other hand, it is a fact that there is a connection between local foods and organic foods, and region- and product-specific differences need to be considered to better market organic and local products. Traditional foods, produced with an organic food-based approach adapted to local conditions based on ecological processes, biodiversity, and plant cycles, are more nutritious and are increasing in popularity [38]. The production of healthier and tastier organic and traditional products is an important factor in agricultural development, while on the other hand, they affect each other. In addition, traditional foods create environments for innovation and quality of life, and combine authentic and traditional flavours and aromas with organic qualities.

## 4. Nutritional Value, Food Quality and Safety

Food security strategies focus not only on the quantity but also on the quality of food. The importance of food quality and safety is increasing day by day in organic farming systems, which are predicted to produce higher nutrient content and quality than conventional systems. Organic farming practices improve food quality and human health, as well as food safety [4]. Foods produced in organic and conventional systems are often compared in terms of nutritional value, sensory quality and food safety [39,40]. The elements most often used in defining food characteristics include functional, natural, sensorial, nutritional, biological and ethical aspects, and authenticity. Food quality is characterised by its nutritional quality, meaning the natural nutritional, biological or health value of a product containing the ratio of beneficial to harmful substances. Food quality can also be described by product, process and consumer-oriented parameters. In most cases, however, except for the first two of these three approaches, consumer quality perception is based on subjective evaluations rather than objective information, such as origin, taste and appearance. While the analytical criteria of food quality include technologically oriented, nutritionally known and sensory valued factors, the holistic criteria, such as in traditional foods, cover authenticity, biological value, ethical aspects and holistic methods of food quality assessment. It has been found that the greatest advantage of organic production is tolerance to water and disease stress; vitamin C is high in organic green peppers and the antioxidant content is higher in conventionally grown produce under no-stress conditions and in organically grown produce under drought conditions [41].

Generally, product quality consists of nutritional values, and sensory, mechanical and functional properties. Nutritional values may be interpreted as vitamins, mineral elements and proteins. In fact, organic products are good alternatives to nutritional supplementation and their nutritional values are slightly higher compared to conventional ones [40,42]. On the other hand, Navarro et al. [43] reported slightly higher nutritional and sensory qualities in organic mandarins than in conventional mandarins; the same was observed by Sreedevi and Divakar [44] who found that the health-promoting nutrients, total soluble solids and sensory qualities in organic ripe bananas were higher than in conventional ones. Indeed, it has been emphasised that the organic management of tea can improve the quality characteristics of tea, thereby providing benefits for human health and the environment [45]. 

Food safety is as important as food quality for the consumer and more eco-friendly organic farming contributes to ensuring food safety in many ways. Organic agriculture systems are the most ancient and widespread practice of sustainable farming and are certainly safer for the environment, although studies on organic foods are fragmented and contradictory, it is clear that they contain fewer pesticides and have better antioxidant properties [46]. However, there is increasing concern about the dependence of agricultural food production on mineral fertilisers and synthetic chemical pesticides due to the reduction in the sustainability of production systems and their negative impact on the environment [47]. The excessive use of chemical fertilisers and other chemicals can lead to soil-quality and -health deterioration, and food-safety and -quality issues such as nitrate build-up in crops and contamination with pesticides and chemicals [1], whereas organic farming can be a good alternative to ensure food safety by reducing the heavy metal content of foods and the negative effects of these chemicals [47,48].

The adverse effects of pollutants on crop quality threaten food safety. In an investigation of 22 pesticides in four different crops (lettuce, apple, grape and tomato), the pesticides levels of samples taken from conventional agriculture were found to be significantly higher than those in organic agricultural products [49]. In terms of safety, there seems to be a consensus that organically grown fruits and vegetables have lower pesticide residue, heavy metal content and nitrate levels, with clear differences in terms of quality and safety between conventional alternatives [48,50,51]. Lower or no pesticide residues in organically produced foods have also been reported in other studies [18,30]. In addition, it has been reported that for the most-consumed vegetables such as potatoes, lettuce, tomatoes, carrots and onions in the US, metal content in conventional products is slightly higher than in organic products [52].

Today, consumer awareness of the impact of the place of origin and method of production on the quality and safety of food, and especially fresh products, is increasing. While it is a fact that available data on quality and safety offer few clear answers, more data are needed to advance knowledge on the safety, health benefits and nutritional quality of organic foods compared to traditional foods. Comprehensive research is required to objectively reflect the differences in nutritional quality and food safety between organic and traditional products [51]. It is seen that factors such as the cultivar, environmental effects and growing conditions, the type of fertilization, the harvest time of the product, the harvest method, storage, transportation and processing techniques are very important for different nutritional, safety and sensory qualities of the product.

## 5. Content of Nutrients, Dry Matter, Vitamins and Other Substances in Crops

A number of previous studies have revealed that organically produced foodstuffs have a higher content of nutrients [50,53,54] and aroma compounds [55]. In general, most organically produced crops have higher dry matter, sugar content, titratable acidity, protective substances, antioxidant potential, flavonoid and total phenol levels, and content of element such as Ca, Mg, P, Zn and Fe than conventionally produced crops. Organic pomegranate juices were found to have higher amounts of acetic acid, alanine, arginine, histidine, glutamine, fumaric acid, lactic acid, isoleucine, leucine, malic acid, galactose, mannose, methionine, phenylalanine, threonine, tyrosine, proline, sucrose, valine and trigonelline than their conventionally grown counterparts [56]. Butternut squashes grown in a conventional system contained higher folic acid and β-carotene, while organic squashes were found to have a higher content of tocopherol, K, Mg, Na and Mn [57]. Organically grown parsley root, celery and potatoes showed higher Ca, Mg, Na, K and P content compared with conventionally grown ones [58]. In addition, little difference was found in organic and conventionally grown greenhouse tomatoes in terms of taste and nutritional value, and regarding the fruit quality index based on the content of compounds such as lycopene, β-carotene and vitamin C [59]. A number of comparative studies show that there is a high ratio of content in organic crop products with more vitamin C and Fe, Mg, Zn, Cu and P than conventional crops [51,60]. Meanwhile, vitamin C is equal or higher in organic potatoes [61], and starch is higher than in conventional ones [62]. Many studies have investigated micronutrient levels in organic and conventional products, due to their quality and nutritional parameters (Table 1).

Vitamin C level was found to be higher in organically grown fruits and vegetables such as peaches, guava fruits, kiwifruit, oranges, strawberries, asparagus, tomatoes, peppers, carrots and mandarins than those that were conventionally produced [18,43,60,84,94,102,110]. In contrast, a study that conducted an organic–conventional comparison [115] reported that L-ascorbic acid content was significantly lower and total soluble solids higher in organic systems compared with conventional ones. Higher carotenoid content was found in organically grown plums, jujube fruit, oranges, mandarins, strawberries, bell pepper, sweet peppers, tomatoes and carrots [43,84,85,102,105], whereas other studies [80,92] found lower or similar content of carotenoids in organically grown green bean, peppermint, lemon balm, sage and rosemary.

On the other hand, results that have been obtained over the past 60 years on the nutrient content of fresh fruits and vegetables grown conventionally in the USA and the United Kingdom have shown a decline in the terms of minerals such as Ca, Mg, Na, K, P and Fe [50,116]. When organically grown, Mg and K in mango, Mo and Al in persimmon, and Cu and Zn in strawberries were high [117]. It has been noted that dry matter, which is an important indicator in the measurement of organic matter accumulation and nutritional composition, is found in higher amounts in organic fruits and vegetables than in conventional ones [15,51,53,92,97,110]. The dry matter content in organic strawberries [97] and the total sugar content in carrots [118] was found to be higher when compared with conventional ones. Generally, organic berries and fruits have high dry matter, vitamin C and antioxidant activity.

## 6. Secondary Metabolites and Antioxidants

Secondary metabolites have great potential to enhance human health. Vegetables and fruits are sources of many beneficial compounds such as polyphenols. Fruits grown in an organic orchard system were found to have higher polyphenol and antioxidant capacity [27,119], while organic pumpkin fruits were reported to contain higher dry matter, total carotenoids, phenolic acids, flavonoids and polyphenols compared to conventional ones [120]. Higher content of secondary metabolites and bioactive compounds and lower content of unhealthy substances such as synthetic fertilisers and pesticides in organically grown compared with conventionally grown food products have been observed in most studies [28,46,51,53]. Moreover, organic cropping systems reported higher results in 11 out of 16 observations of the bioactive compounds in leafy vegetables and fruit crops such as lettuce, cabbage, fennel, tomato, eggplant and apple compared to integrated agriculture [121]. Similar conclusions were presented by Kazimierczak et al. [92], who showed a higher content of beneficial bioactive compounds in organic medicinal and aromatic plants including rosemary, lemon balm, peppermint and sage than in conventional ones.

Plant-derived foods contain natural antioxidants such as flavonoids, polyphenolics, carotenoids and vitamin C, which have been associated with health benefits [122]. Organic fruits and vegetables generally have higher levels of vitamins, dry matter, bioactive compounds, flavonoids, anthocyanins, antioxidants and polyphenolic compounds than conventional ones [27,51,81,85,123,124,125]. Similarly, organic medicinal aromatic *Sideritis perfoliata* had higher levels of secondary metabolites, including total phenolics, flavonoids, vitamin C and antioxidants, compared to conventional ones [126]. Different reports have shown that blueberry fruit grown from an organic culture contained significantly higher total phenolic and total anthocyanins [27]. However, Anjos et al. [127] showed that the phytochemical composition and phenolic compounds of raspberry cultivars had different responses to the same edaphoclimatic conditions—with distinct responses occurring not only between the agricultural practices but also between cultivars—and cannot be generalised. Similarly, organic beetroot samples were found to contain significantly higher total polyphenols and antioxidants than their conventional counterparts; however, the effect of the production system proved to be dependent on the cultivar evaluated [128].

Koureh et al. [124] reported higher values of antioxidant activity, total phenolics, total flavonoids and valuable phenolic compounds in organic-grown white seedless grapes, while D’Evoli et al. [110] found that yield was higher in conventional kiwifruit, but ascorbic acid, antioxidant activity, lutein, β-carotene, total phenol and soluble solid content, and firmness were higher in organic-grown kiwifruit. Organically produced raspberries [123] exhibited higher antioxidant capacities and higher flavonoid, phenolic and soluble solid content compared to those that were conventionally produced. A recent study found that grape berries obtained from organic orchards had better edible quality, are sweeter and softer, have more desired colour parameters, and have higher antioxidant capacity and phytochemical content compared to berries obtained from a conventional orchard [129]. Organically grown green beans [80], table grapes [130], apples [119], tea [45], coffee [95], eggplant [131], legumes [81], wheat [132], rice [54], lettuce [133], beetroot [128] and organic potato and onions [62,69,70] had higher antioxidant capacity than their conventional counterparts. Similarly, in organic grapes, antioxidant-related compounds were significantly higher than in conventionally grown grapes [134].

A number of studies have shown that the content of phenolic compounds is higher in organic products [22,126]. Średnicka-Tober et al. [119], in their study on three apple cultivars from organic and conventional production, found higher content of phenolic acids and the analysed flavonols in organically cultivated apples compared to conventional apples. A number of studies have shown that the content of phenolic compounds is higher in organic products such as apples [119], orange juice [135], pomegranate juice [56], apricots [109], raspberries [123], strawberries [102], barley [136], asparagus [94], onion [69,70], tomatoes [59], potato [61], green pepper, carrot, lettuce [118], bell peppers [85], eggplant [131], tea [45], coffee beans [125], extra-virgin olive oil [137] and table grapes [130]. Similarly, Ribes-Moya et al. [138] found more total flavonoids and luteolin in organic pepper during ripening. However, Guilherme et al. [139] observed that individual phenolic and antioxidant compounds, which vary depending on the cultivar, production system, maturation and climatic conditions, may be higher in conventional peppers in some cases. The stable isotopic δ^15^N value was found to be higher in organic banana pulps than conventional ones, and it was emphasised that this would contribute to our understanding of the compositional differences of bananas due to differences in stable isotope ratios and the elemental composition between production methods [140].

The following reasons have been suggested for the high levels of antioxidants and bioactive compounds such as phenolic acid and flavonoids in organic systems: low N levels in plants grown in organic manure [141], the reaction to various stress factors, limited nitrogen availability [41,120], the activation of plant defence and secondary metabolism under stress conditions [41], the enhancement of protective chemical synthesis in harsh conditions [100], the use of different soil management practices [70], more balanced nutrient management [58], different fertilization practices, and higher soil quality and microbial activity [8,11,12]. Based on the findings that phenolic acid and flavonoid concentrations in wheat grain decrease with mineral fertilization and increase with organic fertilization, it was suggested that phenolics can be optimised by switching to organic fertilisers and reducing or regulating the use of mineral N [47]. Part of these relative differences in antioxidant and phenolic content found between organic and conventionally grown plants can be explained by differences in the nutrient inputs, especially nitrogen, received by the two systems.

## 7. Food Safety

Food safety, which has a significant impact on the food markets and has social importance because it protects human health [142], is the assurance that food will not harm the consumer when prepared or eaten. While the impact of agri-food systems on nutrition and food safety as part of the supply chain is controversial, organic foods are attracting more and more consumer interest [46,51]. In general, it has been found that as age, education and income increase, people pay attention to food safety and quality, and it will also increase their desire to consume organic food due to food safety concerns [143]. Intensive agriculture results in a reduction in the nutritional quality of food and in the sustainability of food production, and organic production methods improve food safety as well as food quality [4]. In this regard, the organic sector aims to minimise the contamination of organic products with synthetic pesticides, which is the main reason consumers buy organic products. Previous comprehensive reviews and meta-analyses of organic versus conventional crop composition data reported that levels of pesticide residues, nitrate content and contamination are substantially lower in organic than conventional crops [4].

Conventional crops tended to have more contaminants such as heavy metals, nitrate and pesticides than organic crops [48,144,145,146,147]. In contrast, others have reported no difference between organic and conventionally grown products [148,149]. Additionally, no differences in pathogen risk were identified between organically versus conventionally cultivated tomatoes, although differences in microbiological quality were apparent [150]. The influence of cropping systems on environmental pollutants, nutrition and food safety is controversial (Table 2).

The production of organically grown vegetables greatly added to the objective of achieving food safety [159]. Appropriate organic farming methods have been found to increase soil nutrient cycling and conservation, could be used to maintain good productivity, and promote food safety and security in apple orchards [160]. With increasing organic management time, pesticide residues in the soil decreased, while organic soils were reported to have fewer residues than conventional soils [161].

Although the effects of organic foods on human health are controversial, they help to reduce food safety risks such as those associated with pesticides and excessive additives. However, organic operators have means to reduce the risk of pesticide residues, but they cannot eliminate all contamination risks [161]; moreover, long-term pesticide contamination of soils may affect existing crops, including organically grown crops [162]. There is strong consensus and scientific evidence that although organic systems have a productivity gap, they provide greater social benefits and ecosystem services [163] and, in terms of food security, organic fruits and vegetables have lower levels of pesticide residues and nitrates [164]. Although, due to the limited database, it is difficult to conclude that one of the production systems is superior and safer compared to the other, it can be argued that organic foods are healthier in terms of chemical hazards, which are food safety indicators.

## 8. Conclusions

This review compared organic and conventional production in terms of quality and nutritional parameters. However, it has been emphasised that the two growing systems should consider other important factors such as organoleptic properties, microbiological safety, social aspects, environmental impact and sustainability. Studies that compare the nutritional values or quality of organic and conventional food products are increasing. The majority of the available scientific literature on plant-derived organic products and their content indicates that they generally contain smaller amounts of nitrate, nitrite and pesticide residues, antibiotics, food additives, industrial pollutants and heavy metals, but higher or equal amounts of mineral elements, vitamins, secondary metabolites, phenolic compounds, antioxidants, anthocyanins, isoflavones, carotenoids, dry matter, total sugars and antioxidant activity, and higher protein quality. Although the reviewed articles have shown differences between organic and conventional foods in favour of organic ones, information is still limited and more research is needed to draw conclusive conclusions. While a few studies have evaluated organic and conventional crop management practices side by side regarding the quality of vegetables and fruits, but it is clear that some results from these studies are contradictory and sometimes inconclusive.

While relatively low nitrogen in organic production systems limits plant nitrogen, fertilization increases protein, nitrogen compounds and the protein–carbohydrate ratio in leafy green, root, tuber and nitrophilic vegetables. Organic farming is more profitable and environmentally friendly, provides ecosystem and social benefits and is equally or more nutritious; however, it will continue to be a minor alternative to conventional farming due to its low yields and cost differential. The requirement to improve the quantity and quality of available food in a sustainable way should orient research on organic agriculture management and food-processing practices toward the use of natural resources, improve the potential of mitigating climate change through organic agriculture, and enhance the nutritional and organoleptic value of agricultural products. With the human population increase requiring higher food production, the organic system should be able to: increase yields by managing local resources without having to rely on external inputs, choose varieties suitable for organic agriculture, develop sustainable strategies to reduce environmental impacts, and minimise threats to biodiversity.

## Figures and Tables

**Table 1 foods-12-00351-t001:** Comparison of the quality and nutritional parameters in organic and conventional fruits and vegetables.

Crops Tested (Foodstuff)	Main Effects of Agricultural System (Higher/Lower/Similar Content in Organic Fruits or Vegetables compared to Conventional)	Ref.
Tomatoes	Higher nutritional value, vitamin C and total flavonoid content, 3-quercetin rutinoside and myricetin in org	[63]
Tomatoes	Higher vitamin C, soluble solids and total phenolics in org	[64]
Tomatoes	Richer health-promoting nutrients, lycopene, vitamin C, flavonoids and total phenolic content in org	[59]
Tomatoes	Higher content of polyphenols in org	[65]
Tomatoes	Higher Mo, Cu, Zn, K and Ba content and lower Mn, Co, Na, Mg and Cd in org	[66]
Tomatoes	Higher content of caffeic acid and chlorogenic acid, but lower ferulic acid and naringenin org	[67]
Tomatoes, ketchups	Higher content of total phenols and antioxidant microconstituents in org tomatoes and tomato-based ketchups	[68]
Welsh onion	Overall, no difference in weight, length, diameter and moisture content; higher total phenolic and flavonoid content and better compositional quality in org	[69]
Red onion	Overall, no difference in individual anthocyanins; higher total phenolic and flavonoid content and antioxidant activity in in org	[70]
Three potato cultivars	Higher nutritional value and total phenolic and dry matter, and better sensory performance, but lower nitrate content in org	[71]
Five potato cultivars	Content of phenolic acids, dry matter and starch, and sensory properties similar in org and conv	[72]
Six potato cultivars	Lower nitrate content; higher nutritional value, total phenolic content and more attractive colour of both the skin and flesh in org tubers	[62]
Sweet potato	Higher concentrations of minerals such as Ca, Cu, Fe, K, Mg, Mn and P in org	[42]
Courgette	No difference in vitamin C, carotenoids, and chlorophylls, but more sugars and polyphenols (gallic acid, chlorogenic acid, ferulic acid and quercetin-3-O-rutinoside) in org	[73]
Carrots	No difference in eating and sensory quality, and overall higher nitrate content in conv	[74]
Carrots, broccoli and zucchini	Carotenoids higher in org carrot, but higher in conv zucchini and broccoli	[75]
Taro	Higher dry matter, starch, sugars, P, K, Ca and Mg content, and better cornel quality in org	[15]
Broccoli	No differences in polyphenol content in org	[76]
Cauliflower	Higher ascorbic acid, polyphenols, carotenoids, and antioxidants in org	[77]
White cabbage	Lower nitrates and nitrites, and higher dry matter, zeaxanthin and β-carotene in org	[53]
Eggplant	Higher nutritional value (K, Ca, Mg, Cu) and total phenolics, but lower polyphenol oxidase activity in org	[78]
Dwarf French bean	No difference in organic acids such as malic, citric and ascorbic acid; higher ascorbic acid, sucrose content and total sugars in org	[79]
Green beans	No difference in carotenoid and polyamines; lower chlorophyll and total phenolics, but higher flavonoid and antioxidant capacity in org	[80]
10 legume cultivars	Higher phenolic acids (namely gallic acid, caffeic acid, syringic acid and ferulic acid) and antioxidant capacity in org	[81]
27 spinach varieties	Lower levels of nitrates and higher levels of flavonoids and ascorbic acid in org	[82]
Lettuce	Higher values of ash, protein, total phenolic compounds and flavonoids, and antioxidant activity, in org	[83]
Sweet pepper	Higher content of sugar, ascorbic acid and yellow carotenoids, and Folin–Ciocalteu index, in org	[84]
Sweet bell pepper	More flavonoids, including myricetin, quercetin, kaempferol, apigenin and carotenoids such as beta-carotene, alpha-carotene, capsorubin and cryptoflavin in org	[85]
Chili fruits	Higher ascorbic acid and capsaicinoid content in org	[86]
Foxtail millet	Higher fructose and glucose content in org	[87]
Rice	No significant differences found for K, Cu, Zn, Rb, Mo or Cd in org	[88]
Rice and wheat	Lower protein, essential amino acid and heavy metal (Cr, As, Cd and Cu) content, but higher flavonoids in both org rice and wheat	[30]
Winter wheat	Lower protein content and levels of Ca, Mn and Fe, as well as toxic elements (i.e., Al, As, Cd and Pb), but higher levels of K, Zn, Mo and quality proteins in org wheat flours	[89]
Winter wheat and spring wheat	No differences in protein content of whole wheat and refined flours, but phenolic content and total antioxidant capacity tended to be lower in org	[32]
Winter wheat	Overall, no difference in total amounts of phenolics and phenolic acid; lower yield, flour proteins and bread-making quality in org wheat	[90]
Buckwheat	Higher amounts of rutin and phenolics in org	[91]
Tea	Higher polyphenols, catechins and the amino acid proline in org	[45]
Peppermint, rosemary, lemon balm and sage	Higher dry matter, vitamin C, phenolic acids and total flavonoids, but lower carotenoids in org medicinal plants	[92]
Sweet basil	Higher concentrations of almost all the major and health compounds in org	[93]
Asparagus	Higher total phenolic compounds, total flavonoids, rutin, vitamin C, chlorophylls, carotenoids and total antioxidants in org	[94]
Coffee	Higher bioactive compound concentration and antioxidants in org	[95]
Strawberries	Higher activities of antioxidant enzymes, and higher antioxidant and flavonoid content in org	[96]
Strawberries	Higher values of dry and optical residue and content of glucose, sucrose, vitamin C and ß-carotene but lower nitrate in org	[97]
Strawberries	No differences in total titratable acidity, lipids, anthocyanins, phenolic compounds, antioxidant activity and vitamin C.	[98]
Red raspberries	Higher values of antioxidant capacities and antioxidant enzymes, and higher anthocyanin and individual flavonoid content in org	[99]
Raspberry	More organic acids in org, but higher vitamin C content in conv	[100]
Blackberries	Higher phenolic-linked antioxidant and anti-hyperglycaemic properties in org	[101]
Goji berry fruits	Higher ash and lipid content and lower proteins, total sugars and total fibres in org	[48]
Oranges, strawberries	Ascorbic acid and β-carotene content higher in org oranges and strawberries, but total phenol content higher in conv oranges and in org strawberries	[102]
Blue honeysuckle berries	Higher total polyphenol and dry matter content in org	[103]
Black Chokeberry	Higher content of bioactive ingredients and antioxidant activity in org	[104]
Jujube fruits	Higher content of chlorophylls, carotenoids, sugars, organic acids and total volatile compounds, and more intense yellow and red colour, but lower protein and flavonoids in org	[105]
*Citrus sinensis*	No differences in total phenolic compounds, vanillic, p-coumaric and ferulic acids; higher hesperidin, total fatty acids and sugar, and lower antioxidant and titratable acidity in org	[106]
Apples	No differences in fruit flesh firmness, sugar content and dry matter, and higher phytochemical concentration, antioxidant capacity, chlorogenic acid, flavonols, flavanols and dihydrochalcones in org	[107]
Juices of fruits (pear, apple and blackcurrant)	Higher content of Ca, Mg, P, Na, Zn, Cu, B, Cd and Ni, but lower S, Na, Cu, B and Ni in org apple juices	[108]
Apricots	More biologically active compound polyphenols and carotenoids in org	[109]
Kiwifruit	Higher fruit performance (flesh firmness, dry matter and soluble solids), antioxidant activity, ascorbic acid, lutein and β-carotene content, but lower yield in org	[110]
Hazelnut	No differences in nut length and thickness, internal cavity, kernel percentage and good kernels	[111]
Ripe banana	No differences in shelf life, and higher sensory qualities (colour, texture and taste) and nutritional qualities (moisture and minerals) in org	[44]
Red grapes	No differences in berry weight, soluble solids, phenolic compounds antioxidant activity and flavonols, and higher anthocyanin and hydroxycinnamic acids in org grapes. Additionally, higher phenolic compounds and antioxidant activity in org wine	[112]
Grape juices *Vitis labrusca*, *V. vinifera*, *V. rotundifolia*	Functional properties, especially antioxidant effects and total phenolic content of org and conv grape juices, were similar; however, higher content of bioactive compounds in org juices	[113]
Grape juices	No differences in phenolic profile, antioxidant activity, and Cu, Fe and Mn minerals between org and conv juices and wines.	[114]

org: organic; conv: conventional; ref: references.

**Table 2 foods-12-00351-t002:** Summary of some studies on the occurrence of contaminants in organic and conventional foodstuffs.

Contaminant	Foodstuff	Remarks	Ref.
Heavy metals (Pb, Cd, Zn, Ni and Cr)	Carrot	Lower quantities of Pb, Cd and Zn in org, but no case exceeded the legal values	[144]
Nitrate content and elemental composition (Na, K, Ca, S, Al, Mg, B, Fe, Zn, Mn, As, Cd, Cr, Cu, Ni and Pb)	Carrot	No differences between conv- and org-grown carrots, and no potential harm arising from heavy metal contamination	[148]
Pesticides and nitrates	Carrot	No pesticide residues and lower content of nitrate in org	[18]
Cadmium (Cd)	Lettuce and carrot	Higher concentrations of Cd in both conv lettuce and carrot, but lower than that established by legislation	[151]
Elemental composition (Zn, Pb, Cu, Cr, Ni, Co and Cd) and nitrates	Tomato	No difference in amounts of Cd, Co and Cr levels, and lower Zn, Pb, Cu, Ni and nitrate content in org	[152]
Elemental composition (Cd, Co, Cr, Cu, Zn, Fe, Mn, Ni and Pb)	Vegetables	Higher concentrations of some elements in conv-grown vegetables; however, the results are not conclusive	[145]
Metal concentrations (As, Cd, Pb, Cr, Ba, Co, Ni, Cu and Zn)	Potato, lettuce, tomato, carrot and onion	All vegetables contained metals, while there were lower concentrations of As, Cd, Pb, Cr and Ba in five org vegetables	[52]
Micronutrients and heavy metals (Ca, Mg, Fe, Mn, Na, Zn, Cu, Ni and Cd)	Vegetables	Decrease in micronutrients in the edible portion of org crops, but increase in toxic metal loads in conv crops	[146]
22 pesticides	Lettuce, apples, grapes and tomatoes	Higher proportion of pesticide levels in conv (9.7%) than in org (2.0%)	[49]
Mycotoxins	Cereals and cereal-based products	No differences in mycotoxin levels between org and conv	[149]
Mycotoxins (deoxynivalenol (DON) and zearalenone (ZEN))	Wheat	DON and ZEN content of org wheat was found to be either lower than or comparable to conv wheat	[153]
Mycotoxins (deoxynivalenol and zearalenone)	Cereal and cereal product	Mycotoxins in org cereals and cereal products did not statistically differ from their conv counterparts.	[154]
Heavy metals (Cd, Hg and Pb)	Goji berries	Lower levels of heavy metals in org	[48]
Mycotoxigenic black *Aspergilli* population	Vineyards	Higher mycotoxigenic *Aspergillus* strains in conv vineyards, and higher risk of mycotoxins in wine originating from these vineyards	[155]
Pesticides, metals, sulphites and ochratoxin A	Wines	No difference in the content of sulphite or ochratoxin, but lower Pb and Mg content, total pesticide concentration and average number of pesticides in org wine	[156]
Sulphite content and pesticide residues	Wines	Higher levels of sulphites, and higher numbers and concentrations of pesticide residues in conv wines	[157]
Pesticide residues, copper and biogenic amines	Wines	Lower numbers and concentrations of pesticide residues and copper in org, but lower concentration of biogenic amines in both groups of wines	[158]

org: organic; conv: conventional; ref: references.

## Data Availability

No new data were created or analyzed in this study. Data sharing is not applicable to this article.

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
