# Peer review of "Quality and Nutritional Parameters of Food in Agri-Food Production Systems"

_foods, 2023, doi:10.3390/foods12020351_

Round 1
Reviewer 1 Report
The paper presented for review is interesting. The authors should elaborate why this research is necessary and how it contribute to practical application. This article were compared to previous findings, but should be further elaborated. The study was well-designed. Please revise the manuscript and correct your English writing.
Please check the detailed comments in the attachment.

Author Response
RESPONSE TO REVIEWER 1 COMMENTS
Point 1: This manuscript deals with the summary and evaluation of the results from researches comparing the quality of traditionally, organically and conventionally produced foods. This paper presented for review is interesting. The study was well-designed, the text clear and easy to read.
Response 1 Thank you very much for your valuable comments.
Point 2.
I have some comments to the authors as follows:
Major remarks
- Please revise the manuscript and correct your English writing
Response: Thank you for your English corrections (Minor remarks). We made them. Also, corrected. Due to the corrections and additions, line numbers have changed in the revised manuscript.
The objectives of this study are not clear. The authors should elaborate why the problems in this paper is necessary and how it contribute to practical application
Response: The objectives of the study were added to the summary section (L 24-28).
Minor remarks:
All minor revisions have been made and are shown in the revised text. Line numbers have changed due to new additions and corrections.
- Line 15: “of the research” instead of “of the results from researches” (L: 15)
- Line 20: “flavonoids compounds” instead of “flavonoids compounds” (two true L 19)
- Line 34: “food’s nutritional” instead of “food nutritional” (L 42)
- Line 35: “contain much fewer” instead of “contains much less” (L 43)
- Line 41: Remove 1 “and”; organic farming have -> has (changed section L 48-51)
- Line 40-43: Brake the sentence into 2 instead (L: 48-51)
- Line 48: The economy (L: 60)
- Line 51: Climate-smart (L 63)
- Line 53: Long-term (L 65)
- Line 55: The soil -> soil (L 66)
- Line 60: The main criticisms are lower (L 72)
- Line 65: comparison (L 80-83, sentence changed)
- Line 70: the yield (L 85)
- Line 71: On input -> input (L 87)
- Line 79: And other (L: 94)
- Line 85: Domain-specific (L 100)
- Line 89: Mouth feel (L 104)
- Line 94: organically product -> organically produced (L 109)
- Line 99: And produced (L: 114)
- Line 100: Agriculture,-> remove comma (L 115)
- Line 107: That of convetional foods (L 122)
- Line112: The nutritional (L 127)
- Line113: A higher (L 128)
- Line 116: Vitamins (L 131)
- Line 119: And that, (L 134)
- Line 121: Has a clear impact on (L 136)
- Line 124: The protein (L 139)
- Line 132: Local, regional (L 147)
- Line 133: Recipes, processes (L 148-149)
- Line 137: The production process (L 156)
- Line 141: Long-time (L 156)
- Line 142: The perceived (L 157-161 , corrected section)
- Line 146: And belonging (L 157-161, corrected section)
- Lines 142 – 146: Try breaking sentence at Moreover, corrected
- Line 147: Considered as (L: 167)
- Line 149: Related to, trends in (L 169)
- Line 150: The rural (L 171)
- Line 151: Ecosystems, -> ecosystems (L 170)
- Line 153: Diversifies (L 173)
- Line 154: Following (L: 174)
- Line 159: The food (L 180)
- Line 165: In order to -> to (L 185)
- Line 166: Food-based (L 186)
- Line 167: And plant (L 187)
- Line 170: other, (L 190)
- Line 172: An organic (L: 195)
- Line 192: green peppers and (L 211-212)
- Line 197: In fact, , are good alternatives (L 216)
- Line 198: as (L 217)
- Line 206: Consumer and (L 225)
- Line 207: Ensureing (L 226)
- Line 232: Offers (L 251)
- Line 237: And , the harvest (L 256)
- Lines 242, 243: A higher (L 261-262)
- Line 252: Organically (L 271)
- Line 253: The conventionally (L 272)
- Line 256: Overall -> A number of (L 275)
- Line 257: a high ratio of content? (L 276)
- Line 258: nutrients or remove it (L 277)
- Line 259: And starch was higher than in conventional (L 278)
- Line 279: In light of result of the past (L 299)
- Line 283: Strawberries (L 303)
- Line 287: Carrots; the total; (L 307)
- Line 295: conventional (L 315)
- Line 299: Of 16 (L 320)
- Line 318: antioxidants (L 339)
- Line 322: Organic-grown (L 345)
- Line 328: Orchards; are sweeter (L 348)
- Line 336: actually (L 356)
- Line 361: increases (L 382)
- Line 374: Plant-derived (L 441)
- Line 375: That possess
- Line 394: The human (L 462)

Reviewer 2 Report
The review article describes the Quality and Nutritional Parameters of Food in Agri-Food Production Systems. It compares the quality of traditionally, organically, and conventionally produced foods. The topic is relevant and interesting. The paper is well written, clear, and easy to understand. I think the subject is overall interesting. The manuscript is well structured. However, the article needs to be more comprehensive. The material is not enough to comprehensively cover the topic. Furthermore, the number of tables and figures can be increased to support and summarize the data. The safety aspect is a key indicator for comparing different food systems. The safety evaluation of organically and conventionally grown foods should be included to conclude and suggest a better food production system. The grammatical errors need to be checked as well.
Line 15; The present review is the summary and evaluation of the results from researches
Line 17; Please revise it “The production systems analysis showed that organically grown vegetables especially berries and fruit”
Line 35; Processed organic foods, compared with conventional, contain much less synthetic additives.
Line 47-52; Please rewrite and elaborate it
Line 53, 54; Long term organic management has been shown to lead to a significant change in the chemical and biological properties of the soil, and has a long-term positive effect on soil quality, microbial diversity, and richness
Line 64-67; However, in comparison with other farming systems, it concludes that organic farming will result in improved soil structure with higher concentrations of organic matter, greater stability of soil biotic and abiotic properties and higher soil aggregation, so the yield gap between organic and conventional farming may decrease over time.
Line 336; A number of studies have shown that the content of phenolic compounds is higher in organic products.
Author Response
RESPONSE TO REVIEWER 2 COMMENTS
Point 1: The review article describes the Quality and Nutritional Parameters of Food in Agri-Food Production Systems. It compares the quality of traditionally, organically, and conventionally produced foods. The topic is relevant and interesting. The paper is well written, clear, and easy to understand. I think the subject is overall interesting. The manuscript is well structured. However, the article needs to be more comprehensive. The material is not enough to comprehensively cover the topic. Furthermore, the number of tables and figures can be increased to support and summarize the data. The safety aspect is a key indicator for comparing different food systems. The safety evaluation of organically and conventionally grown foods should be included to conclude and suggest a better food production system. The grammatical errors need to be checked as well.
Response 1: Thank you very much for your valuable comments. A section entitled "Food safety" has been added to the article (7. Food Safety (Revised text: L 388-432). A table has also been prepared and attached on the subject (Table 2). Now I think the manuscript is more comprehensive. Thanks again for this suggestion. Added references (142-164 L: 797-846). The grammatical errors have been checked.
Point 2:
All minor revisions have been made and are shown in the revised text. Line numbers have changed due to new additions and corrections.
Line 15; The present review is the summary and evaluation of the results from researches
Response: corrected in the revised text (L 15)
Line 17; Please revise it “The production systems analysis showed that organically grown vegetables especially berries and fruit”
Response: corrected in the revised text (L 17-18)
Line 35; Processed organic foods, compared with conventional, contain much less synthetic additives.
Response: checked ( L 43)
Line 47-52; Please rewrite and elaborate it
Response: corrected (L: 48-51)
Line 53, 54; Long term organic management has been shown to lead to a significant change in the chemical and biological properties of the soil, and has a long-term positive effect on soil quality, microbial diversity, and richness
Response: checked, corrected (L 65-67)
Line 64-67; However, in comparison with other farming systems, it concludes that organic farming will result in improved soil structure with higher concentrations of organic matter, greater stability of soil biotic and abiotic properties and higher soil aggregation, so the yield gap between organic and conventional farming may decrease over time.
Response: changed, corrected (L 80-83)
Line 336; A number of studies have shown that the content of phenolic compounds is higher in organic products.
Response: corrected (L: 359-360)

Round 2
Reviewer 2 Report
The review article describes the Quality and Nutritional Parameters of Food in Agri-Food Production Systems. It compares the quality of traditionally, organically, and conventionally produced foods. The safety espect has been added.